# Design and Simulation of Tunneling Diodes with 2D Insulators for Rectenna Switches

**DOI:** 10.3390/ma17040953

**Published:** 2024-02-19

**Authors:** Evelyn Li, Parameswari Raju, Erhai Zhao

**Affiliations:** 1Thomas Jefferson High School for Science and Technology, Alexandria, VA 22312, USA; evelynli2006et@gmail.com; 2Department of Physics and Astronomy, George Mason University, Fairfax, VA 22030, USA; praju3@gmu.edu

**Keywords:** MIM junction, 2D insulators, Switch, tunneling diodes, rectenna, rectifying circuits

## Abstract

Rectenna is the key component in radio-frequency circuits for receiving and converting electromagnetic waves into direct current. However, it is very challenging for the conventional semiconductor diode switches to rectify high-frequency signals for 6G telecommunication (>100 GHz), medical detection (>THz), and rectenna solar cells (optical frequencies). Such a major challenge can be resolved by replacing the conventional semiconductor diodes with tunneling diodes as the rectenna switches. In this work, metal–insulator–metal (MIM) tunneling diodes based on 2D insulating materials were designed, and their performance was evaluated using a comprehensive simulation approach which includes a density-function theory simulation of 2D insulator materials, the modeling of the electrical characteristics of tunneling diodes, and circuit simulation for rectifiers. It is found that novel 2D insulators such as monolayer TiO_2_ can be obtained by oxidizing sulfur-metal layered materials. The MIM diodes based on such insulators exhibit fast tunneling and excellent current rectifying properties. Such tunneling diodes effectively convert the received high-frequency electromagnetic waves into direct current.

## 1. Introduction

In general, a typical rectenna, i.e., rectifying antenna, consists of an antenna and rectifying circuit. The rectenna was first developed by Brown in 1963 [1]. The rectifying circuit is used to convert electromagnetic (EM) waves into direct current (DC) electricity. The conventional rectifying circuits work properly with the radio-frequency (RF) signals of a range from 3 kHz to 500 kHz. [2] Certain carefully designed rectifying circuits could be operating at up to 300 MHz with bipolar junction transistors and 2 GHz with the integration of multiple MOS devices [3,4]. As telecommunication is entering into the 5G and soon 6G era, both a fast response and a large bandwidth are needed, pushing the limit of frequency toward the high end of the RF range (>100 GHz) [5]. For example, millimeter waves (30–300 GHz) and terahertz signals will be used in 6G technology [6,7]. The 100 GHz signal demands 10 ps in response time, approaching the limit in transit time for an electron to diffuse across a silicon PN junction diode. Such a long diffusion time places a strict limit on the high-frequency application in converting alternative current (AC) into DC.

On the other hand, optical detection, as an important method for non-ionizing medical detections [8,9], has been increasingly used in wearable electronics, e.g., smart watches, to measure heartbeat, blood sugar, blood oxygen, and so on. This technology requires the lights, a kind of high-frequency electromagnetic waves, to penetrate through the skin [10]. The current wearable electronics are often equipped with visible lights, e.g., green and red lights, to perform the biomedical measurement [11]. However, the medical information obtained by such wearable electronics is quite limited and not precise. THz propagation through the biological tissues would be accompanied by much less scattering loss because of its much longer wavelength compared to those of visible or infrared radiation [12]. It is well known that EM waves at 10^12^ Hz (THz) are able to image human tissues without harm and provide much more information at a higher resolution in comparison with other methods. For example, ultrasound, another method for non-ionizing detection, with frequencies ranging from 2 MHz to 12 MHz, has a resolution at 1 mm, much lower than that of optical and THz detection. Thus, THz electronics offer excellent safe evaluation, high resolution and precision, all of which are very attractive for future medical detections [13,14]. Since the optical and THz signal frequencies are too high for conventional semiconductor diodes, new diodes with a switching time that is faster than 1 ps are needed to rectify THz signal for medical detection.

In addition, rectenna has been studied for the energy harvesting of 2.4 GHz EM waves for Internet-of-Things technology [15]. To receive and rectify EM waves at optical frequencies, e.g., solar energy, optical rectennas [16,17] become attractive high-efficiency and low-cost solar cells if fast-switching diodes are available. Conventional solar cells are based on a semiconductor PN junction whereas electron–hole pairs are generated by photons with an energy that is larger or close to the energy bandgap (E_g_) of the semiconductors. Silicon-based solar cells in the most commercially dominated products have a limited efficiency, about 16% for polycrystalline Si, 30% for monocrystalline Si, and 55% for multi-junction cells [18,19,20]. In comparison, the rectenna solar cells have a much higher efficiency (>85%) and can be fabricated at low cost [16].

Therefore, the common technical challenge for the above-mentioned emerging technologies (high-speed telecommunication, THz electronics and rectenna solar cells) hinges on new rectifier diodes. As shown in Figure 1a, in the conventional semiconductor PN junction diode, the excess carriers are injected to each side of the junction as minority carriers and diffuse through the diode. The diffusion time (τ_s_) is close to the smallest value of the electron lifetime (τ_e_) or hole lifetime (τ_h_). The lifetime of a minority carrier depends on the Auger lifetime due to Auger recombination [21]. In typical, for Si, with a doping of 10^18^ cm^−3^, the diffusion time is within a range from 10 ns to 1 μs, which corresponds to switching frequencies from 1 MHz to 100 MHz. The optimization of device structures and diode materials, such as Schottky junction diodes and GaN diodes, has been studied to improve the switching frequency to GHz [22]. However, THz and optical frequencies are still too high for conventional PN junction diodes.

On the other hand, carrier transport based on tunneling can easily exceed the limit of diffusion in PN junctions, reaching the scale of femtoseconds (10^−15^ s). The metal–insulator–metal (MIM) junction with an atomically thin insulator between two different metals is a simple yet very effective tunneling diode. A schematic of the MIM tunneling diode and its energy barrier for tunneling are shown in Figure 1b. The electron tunneling time is determined by the energy barrier profile and applied bias across the barrier. Studies showed that the tunneling time across a tunneling diode could reach femtoseconds [23] and even attoseconds (10^−18^ s) in an atomic hydrogen [24].

The MIM heterostructures have recently been studied as a current rectifying diode [25,26,27,28]. The research interest of MIM diodes was focused on application in rectifying high-frequency electromagnetic waves, THz electronics, infrared light, and even the conversion of visible light into electricity [16,29,30]. The MIM diodes based on bulk insulators, such as TiO_2_, ZnO, and NiO, could be applied in 28.3 THz rectenna switches [31]. The amorphous InGaZn oxide was shown to improve the turn-on voltage control in the diodes [32]. Amorphous metal electrodes could also improve the current rectifying performance [33]. Recently, MIM diodes based on 2D insulator materials have been reported [34]. MIM diodes with single and multiple insulators have been studied and analyzed for the conversion of infrared light into electricity [35,36]. Despite the rapid progress, there still lacks a comprehensive study that combines the first-principle calculation for designing 2D insulator materials, the modeling of tunneling diodes, and the simulation of rectifier circuits.

In this work, we applied a density function theory (DFT) method to design and model MIM tunneling diodes with two-dimensional (2D) materials as the insulators. In the modeling, 1, 2, 3-layer TiO_2_, TaO_2,_ and SnO_2_ were designed by oxidizing the corresponding 2D metal-sulfur materials. The monolayer or few-layer 2D materials were placed between two different metals to form a tunneling junction diode. In the device simulation, it is found that the Au/monolayer TiO_2_/Al tunneling junction diode exhibits the best rectifying performance. The simulated tunneling current–voltage (I–V) characteristics were fed through a rectifier in PSpice modeling, showing excellent signal rectifying from THz to optical frequencies.

## 2. Modeling and Simulation Method for Materials, Devices and Circuits

In this study, the atomistic simulations of 2D materials and MIM tunneling diodes were performed in QuantumWise Atomistix Toolkit (ATK) simulation platform [37,38,39] from a Synopsys based on density functional theory (DFT). The DFT calculations employed the norm-conserving Pseudo-Dojo pseudopotentials [40] with the Perdew–Burke–Ernzerhof (PBE) parametrization of the exchange-correlation function [41] for all materials and diode structures. To bring all atoms to the ground state, a 12 × 12 × 1 Monkhorst-Pack k-point grid mesh was used for the sampling of the Brillouin zone. The forces were kept to the least at 0.01 eV/Å. For the calculations of carrier transport in the tunneling diodes based on metal/2D insulators/metal heterostructures, we used first-principle calculations integrated with density functional non-equilibrium Green’s function (NEGF) method. In addition, the current–voltage (I–V) characteristic was calculated by following the Landauer–Büttiker approach [42]. The design of 2D materials and the computation of carrier transport were implemented using the above-mentioned QuantumWise ATK code [37].

In the simulation of 2D materials, 1-, 2-, 3-layer metal–sulfur materials are designed and modeled with structural relaxation. These ultra-thin 2D metal–sulfur materials are usually semiconductors with an energy bandgap. [34] In the structural relaxation in simulation, the position of all atoms and the three-dimensional lattice parameters were optimized with the least Hellmann–Feynman force of 0.01 eV/Å on the metal–sulfur layer materials. In the optimization, Pulay-mixer algorithm was used as a fully self-consistent field (SCF) iteration control with a tolerance value of 10^−5^ eV. The maximum number of iteration steps was set at a limit of 100. The self-consistent field computations were strictly tracked to guarantee full convergence within the iteration steps. Periodic boundary conditions were employed along the three directions (in-plane and out-of-plane) with a large vacuum region of 7 Å to minimize the interlayer interaction. [43] Energy band structures and the electronic properties of the metal–sulfur layer materials could be calculated once the structure optimization simulation was converged. In the design and simulation of metal–oxide layer materials, we proposed a simulated oxidation process: (1) the sulfur atoms were replaced with oxygen atoms; and (2) the new metal oxide materials were fully optimized by following the similar procedure in obtaining relaxed metal–sulfur layer materials. As observed in the calculation of formation energy (E_Form_), the metal sulfides are more stable than metal oxide layer materials. For example, the E_Form_ of TiS_2_ and TiO_2_ is –1.96 eV and −3.36 eV, respectively.

The 2D layer oxide insulators were placed between two metal electrodes to construct metal/2D insulators/metal tunneling diodes. There is a 7-Å gap between the 2D insulators and electrodes at each side to protect the layer oxides from a chemical reaction with the metal electrodes during simulation. As the layer insulators were surrounded by vacuum, our simulation was able to neglect the effect of substrate and electrodes. The simulation of the tunneling current in the metal/2D insulators/metal diodes under a bias voltage was implemented using DFT method in the Virtual Nanolab ATK package. [37] Generalized gradient approximation (GGA) [41] was adopted with PBE exchange correlation to describe and include the electron correlation and exchange energies in the calculation of carrier transport under bias voltage.

The design and modeling of a full-bridge AC/DC power converter built on the proposed MIM diodes was carried out using OrCAD PSpice simulation tool [44]. In the circuit simulation, a PSpice device model was built for the MIM tunneling diodes by implementing the I–V characteristics and the tunneling time obtained in above-mentioned DFT calculations. An AC voltage input of 1.0 V at 100 kHz was applied in the simulation and a DC output was analyzed to evaluate the performance of proposed MIM tunneling diodes. The output voltage as a function of received signal frequency from 1 Hz to 1000 THz was simulated based on the calculated tunneling time to analyze the high-frequency performance.

## 3. Results and Discussion

### 3.1. Molecular Structures and Energy Bands of 2D Metal Oxide Insulators

Figure 2a shows the design of MIM tunneling diodes with the 2D insulator material sandwiched between the top and bottom metal electrodes. The two electrodes have a different work function, enabling current rectifying functionality of the MIM diodes. The key element of the diodes is the middle 2D insulator and its interface with metal electrodes. It is desirable to use insulators with a low tunneling barrier height, i.e., high electron affinity, such as TiO_2_, SnO_2_, and Ta_2_O_5_, to rectify a low-voltage–current characteristic [33,34]. Lower barrier height generally leads to a higher tunneling rate. However, the efficiency of the tunneling current also depends on the quality of the interface between the insulator and metal electrodes. The proposed 2D insulators have a smooth and self-passivated surface, very favorable for tunneling. As shown in Figure 2b, the 2D layer-structured oxide insulators, such as TiO_2_, SnO_2_, and TaO_2_, can be obtained by oxidizing the counter parts 2D metal sulfur layer materials, such as TiS_2_, SnS_2_, and TaS_2_. These 2D layer materials are van de Waals materials with the monolayer or multilayer, whereas the S atoms can be replaced with O atoms by oxidation. In the simulation, the oxidation is expressed by the replacement of S atoms with O atoms followed by a structural relaxation to achieve stable 2D oxide layer materials.

Figure 3 shows the energy band structures of hexagonal TiS_2_ and TiO_2_ obtained from the DFT simulation. As mentioned above, the 2D TiO_2_ materials are achieved by replacing S atoms in TiS_2_ with O atoms followed by a structural relaxation and optimization. The DFT calculation of 2D oxides converged at a minimal force of 0.01 eV/Å, achieving a stable layer structure which can be obtained in the experiment and in production. After such a proper structural relaxation, the energy band structures and electronic properties of each monolayer, bilayer, and trilayer structures can be reasonable evaluated.

The result indicates that the 1-, 2-, 3-layer TiO_2_ materials have a significant bandgap and can be considered insulators. The energy bandgap value increases from 1.7 eV for the monolayer TiO_2_ to 3.0 eV for trilayer TiO_2_. The 2D TiO_2_ have a significantly larger bandgap than the counterparts of 2D TiS_2_. TiS_2_ is considered a semiconductor with a relatively small energy bandgap. [45,46] The simulated band gap for monolayer materials will be wider if the spin-orbit coupling (SOC) and Heyd–Scuseria–Ernzerhof (HSE) functional are considered in calculation [47,48,49,50]. The SOC downshifts valence band maximum and upshifts conduction band minimum by including the interaction of an electron’s spin with its motion. [49] The HSE adds an additional potential by describing many-electron interactions and charge localization. [50] Nonetheless, the material structure and electrical properties of 2D TiO_2_ are successfully modeled and can be used for the simulation of carrier transport, tunneling diodes, and voltage conversion circuits.

Similar DFT simulation processes are also performed on the TaS_2_ and SnS_2_ layer materials, and the TaO_2_ and SnO_2_ layer materials obtained via the oxidation of the metal–sulfur counterparts. Figure 4 shows the molecular structures of the above-mentioned trilayer materials. The 2D layer materials were designed with thin vacuum region (7 Å) surrounding them to minimize chemical interaction with substrate or electrodes. These 2D TaS_2_, SnS_2_, TaO_2_, and SnO_2_ were then optimized with structural relaxation using the QuantumWise ATK code [37,41]. The energy band structures of TaS_2_ and TaO_2_ mono-, bi-, and trilayer are shown in Figure 5. The result indicates that a significant band gap exists in TaO_2_ layered materials while the band gap is negligible in TaS_2_ layered materials, which has been previously reported [51]. Also, the band gap of 2D TaO_2_ (about 0.7 eV) does not change appreciably with an increasing number of layers, which is different with that of TiO_2_. In addition, the band structures of 2D TaO_2_ materials look like the ones of n-type semiconductors with the Fermi level merged into the conduction band.

Figure 6 shows the energy band structures of mono-, bi-, and trilayer SnS_2_ and SnO_2_ materials. The 2D SnS_2_ layer materials have a significant indirect bandgap about 1.6 eV.

However, the energy band gap of SnO_2_-layered materials disappears after the SnS_2_ is oxidized with the S atoms being replaced by O atoms. From the observation from the band structures, the SnO_2_ layer materials look more like a semimetal with zero bandgap than a semiconductor or insulator. The result indicates that the SnO_2_ layer materials are not a good choice as 2D insulators for the application in the proposed MIM diodes.

### 3.2. Design and Electrical Properties of MIM Diodes with 2D Insulators

As seen from the DFT calculations, although some 2D metal oxide materials are not a good choice for insulators, excellent 2D materials can be screened and found by using such a simulating oxidation experiment based on the DFT calculation. The next step is to design an MIM diode using the modeled 2D insulators and study its electrical properties. As shown in Figure 7, MIM diodes are constructed with monolayer, bilayer, or trilayer TiO_2_ being sandwiched between Au and Al electrodes. The 2D TiO_2_ films act as the smooth, ultra-thin tunneling barrier for the electron to transport between two metal electrodes. In the design of Au/TiO_2_/Al heterojunction, 9 atomic layers of Au and Al have been employed to maintain the structure of metal during the structural relaxation in the simulation. This significantly increases the computation time in the simulation since a large number of atoms are included. However, this design is sufficiently close to the real devices in which metal films are much thicker than 2D insulators and will not experience structural change due to a lattice difference with the 2D materials. It should be noted that the metal structures have an unreal change in structural relaxation if only < 5 atomic layers of metals are used in simulation, whereas the thickness of metal is comparable to the inserted 2D insulators. In addition, the design of MIM tunneling diodes is not limited to 2D TiO_2_ and can be applied to other 2D insulators.

Figure 8 shows the I–V characteristics of Au/TiO_2_/Al tunneling diodes with the monolayer, bilayer, and trilayer of TiO_2_ at room temperature. In the simulation, an electric bias voltage from −1.0 V to 1.0 V is applied on MIM diodes at a step of 0.2 V. The I–V curves exhibit a junction-like current rectifying function with low resistance for the current flowing from the Au electrode to the Al electrode and high resistance along the opposite direction. Although all three MIM diodes have a current-rectifying capability, the current density decreases exponentially with increasing layer numbers of TiO_2_ films. In addition, the I–V curves become less exponential, i.e., the current on/off ratio within a certain voltage window decreases, and the number of TiO_2_ layer increases. This indicates that the rectifying capability decreases with the increasing thickness of TiO_2_ layer materials. The simulation result in this work showed a high current density (J) (J = 500 A/cm^2^) at 1.0 V. In comparison, the experimental result [34] of the layer TiO_2_ MIM diode (5 nm layer TiO_2_) showed J = 200 A/cm^2^ at 2.0 V, and the experimental result of Pt/TiO_2_/TiO_2-x_/Ti diodes (3 nm TiO_2_) [36] showed J = 350 A/cm^2^ at 1.0 V. Although our simulated devices showed a higher on-state current density, the off-state current density at reverse bias (−1.0 V) is about 80 A/cm^2^, higher than that of the layer TiO_2_ MIM diodes [34] which showed an off-state current density of 20 A/cm^2^ at—2.0 V and slightly lower than that of Pt/TiO_2_/TiO_2-x_/Ti diodes [36] (about 150 A/cm^2^ at reverse bias—1.0 V). The high off-state current is due to the thin TiO_2_ monolayer in simulation.

In addition to the selected Au and Al electrodes, combinations of several metals with a work function from 4.1 eV to 5.6 eV, including Al, Ti, Ni, Cu, Ag, Au, and Pt, have been tried in the simulation. Our study shows that the Au/monolayer TiO_2_/Al diode exhibits the best current rectifying characteristics: a large rectifying ratio, low off-state current and a high on-state current. In the circuit modeling (to be discussed later), we selected the Au/monolayer TiO_2_/Al tunneling diode as the switching diode model for voltage conversion and adjusted the current level or diode resistance for application by choosing the right cross-section area. It should be noted that the tunneling time is not be affected by the device area.

The tunneling time is an important parameter for rectifier diodes. It directly determines the upper limit of the switching frequency that the rectifier can handle. In this work, the tunneling time of the Au/monolayer TiO_2_/Al diode have been studied and calculated for different bias voltages. Figure 9 shows the electron affinity extracted from the simulation for the monolayer TiO_2_ between Au and Al electrodes. A trapezoid shape of the tunneling barrier is drawn along the extract value of electron affinity for each bias voltage (see Figure 9a–c). The trapezoid barrier profile will be used for the calculation of the tunneling time. The tunneling time can be calculated using the integration equation shown in Figure 1b following the method by Buttiker and Sinton [52,53]. As shown in Figure 9d, the tunneling time slightly decreases from 1.85 fs to 1.65 fs with increasing bias from 0 V to 1.0 V. Such a short switching time enables the proposed diode to effectively the rectify high-frequency signal up to 600 THz. It should be noted that the simulation does not consider the impact of impurity and defect at the interface and inside the 2D materials. The interface impurity and defect usually increase the tunneling time and decrease the tunneling current, which should be an important issue to examine in the experiment. Nonetheless, the proposed metal/2D insulator/metal tunneling diodes exhibit an excellent current rectifying capability with an ultra-fast switching time reaching the order of femtoseconds (10^−15^ s).

### 3.3. Circuit Simulation of Rectifiers Based on 2D MIM Tunneling Diodes

The performance of 2D MIM tunneling diodes can be fully evaluated in a rectifying circuit. Figure 10a shows a schematic of the rectenna circuit with both an antenna and a full-wave bridge rectifying circuit. In the rectifier circuit, D1, D2, D3, and D4 are identical diodes based on the above-simulated Au/monolayer TiO_2_/Al MIM tunneling diode with the I–V characteristics shown in Figure 8 and the tunneling time (1.6 fs) derived in Figure 9d. The diodes have a cross-section area of 10^−4^ cm^2^ (i.e., 100 μm × 100 μm), exhibiting an on-state current at the range of 10 mA–40 mA. The circuit simulation was carried out and analyzed in OrCAD PSpice Integrated Simulation platform [44]. In the circuit design, the antenna is replaced with an AC voltage source which is tunable in voltage and frequency. Both the AC-to-DC conversion and high-frequency performance were studied and analyzed. Figure 10b shows that the rectifying circuit effectively converts a 1.0V, 100kHz AC input, simulated the signal received by the antenna, into a 0.7V DC voltage output. The DC output voltage is stabilized within 20 ms, exhibiting a robust AC/DC conversion with the proposed 2D MIM tunneling diodes. Figure 10c shows the DC output voltage as a function of the frequency of input AC signals in the circuit simulation. The rectifying circuit based on the proposed monolayer TiO_2_ MIM tunneling diode successfully maintained a DC output voltage of about 0.7 V for up to 800 THz, agreeing with the switching time of the tunneling diodes. It should be noted that the output voltage is stable in the visible region from red (430 THz) to blue light (750 THz), which is very attractive for the application in rectenna solar cells. Such a THz operation frequency range is also important for medical detection and high-speed telecommunication.

## 4. Conclusions

In summary, we have designed, simulated, and investigated a family of 2D insulators by oxidizing metal sulfides using the DFT method. The I–V characteristics of the metal/insulator/metal junctions with the chosen 2D insulators exhibit an excellent rectifying capability with a significant dependence on the number of layers. The diode tunneling time has also been studied, showing promising potential for THz electronics. In addition, we have successfully designed and simulated an AC-to-DC rectifying circuit based on the proposed monolayer TiO_2_ MIM tunneling diode. The rectifier is able to robustly convert AC voltage input into DC voltage output. The rectifier circuit is effectively functional at a high frequency up to 800 THz. Such rectifying switches based on the proposed 2D insulator MIM tunneling diodes are very attractive in many applications including medical detection, 5G/6G telecommunication, and rectenna solar cells.

## Figures and Tables

**Figure 1 materials-17-00953-f001:**
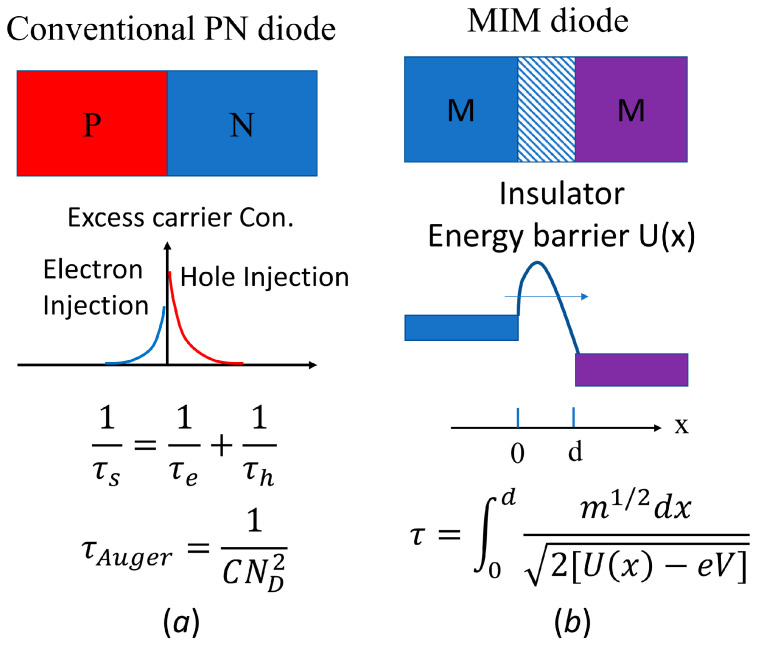
(**a**) In a conventional PN diode, the charge carriers diffuse through the depletion region, limited by the Auger recombination time (τ_Auger_). The diffusion time (τ_s_), which is the sum of the reciprocals of the life time of the electron (τ_e_) and hole (τ_h_) carriers, is usually long, reversely depending on the square of doping concentration (N_D_), where C is a constant. (**b**) In an MIM tunneling junction, the electron tunnels across a thin barrier. The tunneling time (τ), depending on barrier height profile, U(x), barrier thickness (d), applied potential (V), and electron effective mass (m), is usually very short.

**Figure 2 materials-17-00953-f002:**
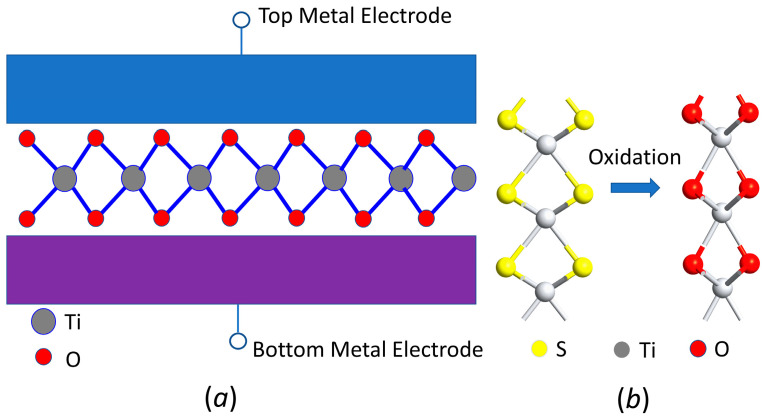
(**a**) Schematic of the MIM junction with 2D layered insulators being sandwiched by top and bottom electrodes. (**b**) The molecular structure of monolayer TiS_2_ and TiO_2_. Two-dimensional metal oxide insulators can be obtained by oxidizing metal sulfides. For example, layer TiS_2_ can be oxidized into layer TiO_2_.

**Figure 3 materials-17-00953-f003:**
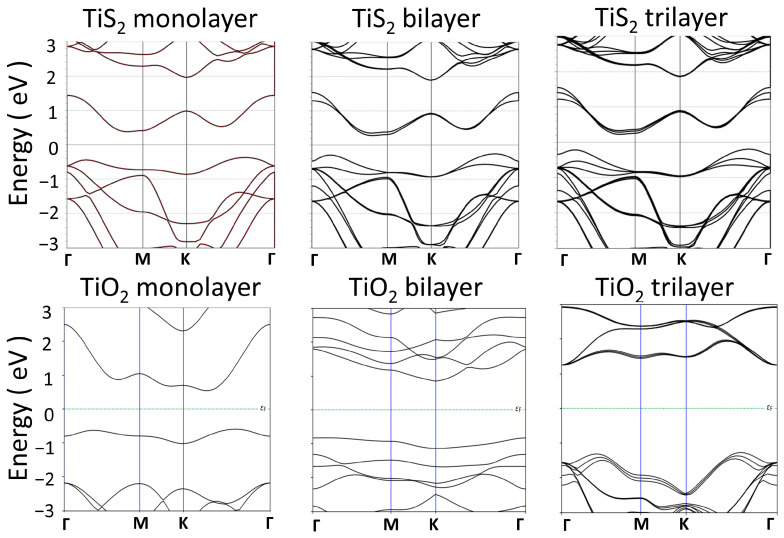
Energy band structures of monolayer, bilayer, and trilayer TiS_2_ and TiO_2_.

**Figure 4 materials-17-00953-f004:**
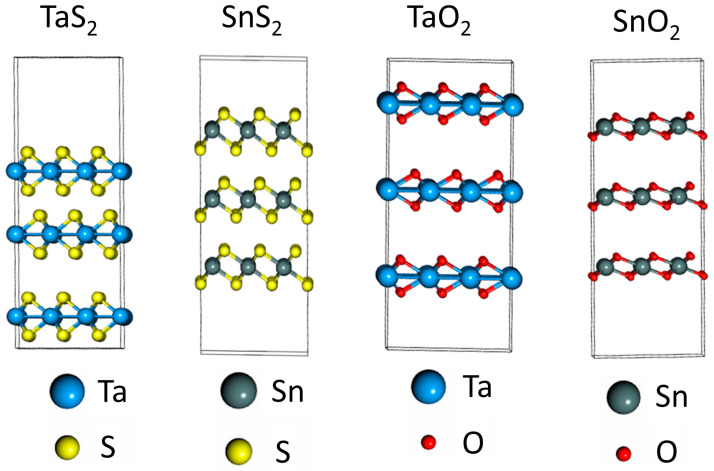
Design of molecular structures for trilayer TaS_2_, SnS_2_, TaO_2_, and SnO_2_ in simulation.

**Figure 5 materials-17-00953-f005:**
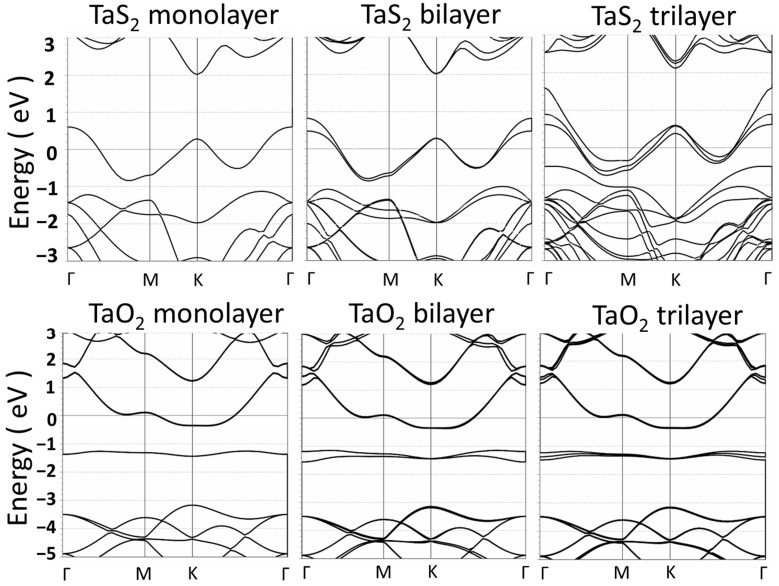
Energy band structures of TaS_2_ and TaO_2_ monolayer, bilayer, and trilayer.

**Figure 6 materials-17-00953-f006:**
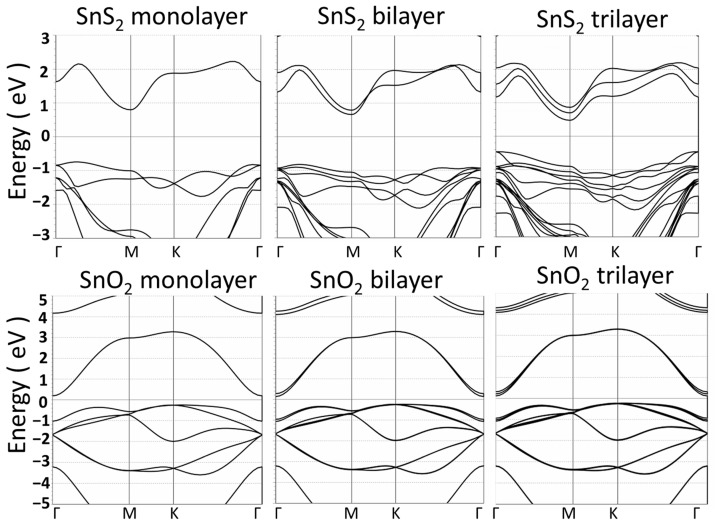
Energy band structures of SnS_2_ and SnO_2_ monolayer, bilayer, and trilayer.

**Figure 7 materials-17-00953-f007:**
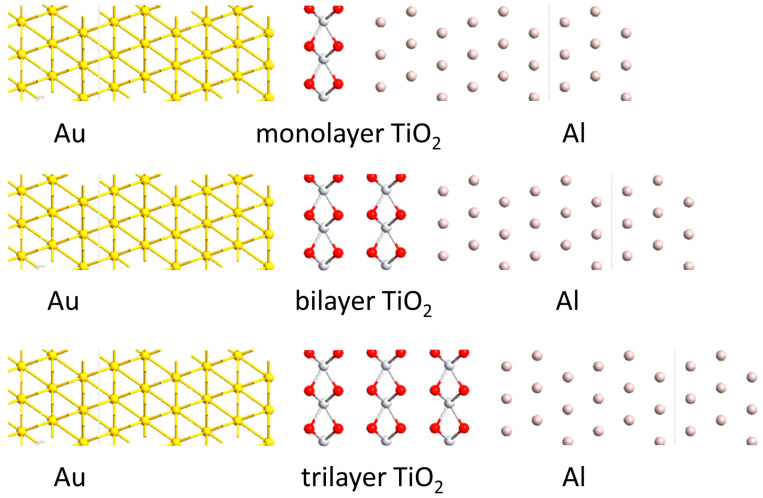
Design of Au/TiO_2_/Al MIM tunneling diodes with the monolayer, bilayer, and trilayer TiO_2_.

**Figure 8 materials-17-00953-f008:**
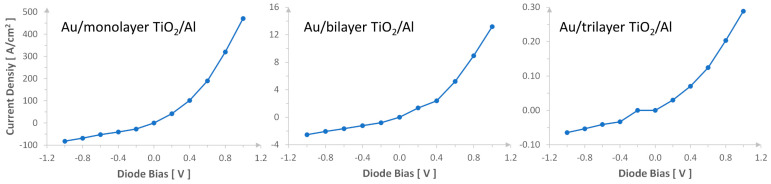
The current–voltage (IV) characteristics for Au/TiO_2_/Al tunneling diodes.

**Figure 9 materials-17-00953-f009:**
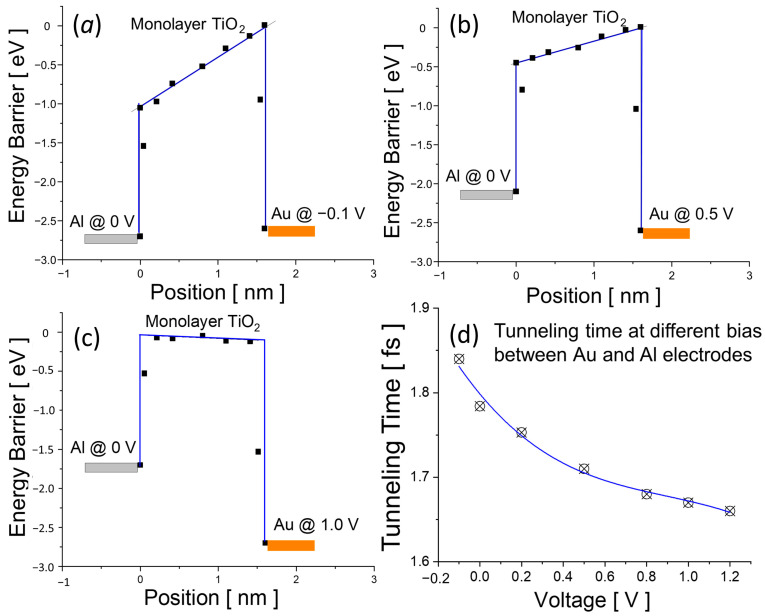
Electron affinity profile of the monolayer TiO_2_ at a position along the current flowing direction is extracted when the diode is biased at (**a**) −0.1 V, (**b**) 0.5 V, and (**c**) 1.0 V. A trapezoid marked in blue is drawn along the electron affinity to represent the tunneling barrier for each case. (**d**) Tunneling time for each bias is calculated according the method described in Figure 1.

**Figure 10 materials-17-00953-f010:**
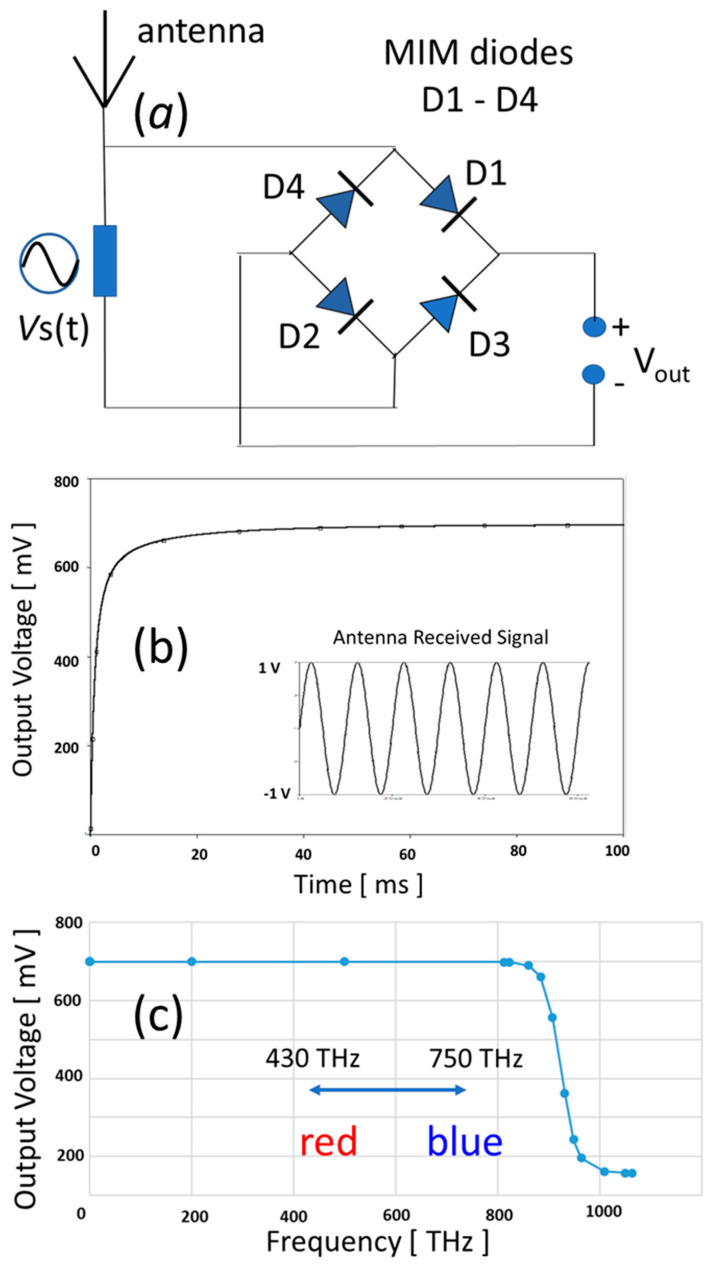
(**a**) Schematic of a rectenna circuit with a full-wave bridge rectifier based on the modeled monolayer TiO_2_ MIM diode. (**b**) Simulation of the rectifier which converts the AC signal into the DC output. (**c**) DC output voltage as a function of input AC frequency. The rectifier maintains a constant DC output at 0.7 V for up to 800 THz. The output voltage is stable in the visible region from red (430 THz) to blue light (750 THz).

## Data Availability

Data available upon request.

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
