# Peer review of "Design and Simulation of Tunneling Diodes with 2D Insulators for Rectenna Switches"

_materials, 2024, doi:10.3390/ma17040953_

Round 1

Reviewer 1 Report

Comments and Suggestions for Authors

The authors reported DFT calculations for 2D materials for selecting suitable tunnel barriers for tunneling diodes and the circuit simulations for the device characteristics of the tunneling diodes. The results point out a way to incorporate 2D materials, especially transition metal dichalcogenides, to design and fabricate tunneling diodes and related devices. This could contribute to the electronic industry. The work was well structured and the results presented are sound. The manuscript is also well-written and comprehensive. However, some points can be explained more clearly for the readers to understand better, and therefore, I suggest minor revision for this report before it can be published.

Comments:

1. The authors use "oxidation and then relaxation" approach to carry out the DFT calculations for 2D Ti-based dichalcogenides. What is the advantage of this approach than "direct calculation" by putting the parameters into the supercells?

2. Why was the spin-orbit coupling (SOC) not considered in the DFT calculation? How does it relate to the underestimate of the band gap?

3. For Figure 10(c), the output voltage versus AC frequency, it is a pity that no data points between 100 kHz and 800 THz. If there were, it would be more convincing. Moreover, what do the words "red" and "blue" represent? Why were the frequency 430 THz and 750 THz highlighted?

Author Response

The response is attached in the document: response to review 1.

Reviewer 2 Report

Comments and Suggestions for Authors

This work comprehensively studies a newly designed metal-insulator-metal (MIM) diode as a promising rectenna to rectify high-frequency signals. TiO2 was identified as the most promising candidate among a family of 2D insulators. The density functional theory (DFT) was utilized to obtain the optimized geometry and electronic properties. From their DFT calculations, TiO2 was identified as the best candidate insulator. Additionally, the tunneling and excellent current rectifying properties were further carried out. Although the interfacial and surface defects, which are key factors in determining tunneling and rectifying properties., were not carried out in this study, this work is still intriguing and has a scientific merit. Therefore, I am in favor of publishing this manuscript after minor revision. My comments for the further improvement of this manuscript are below.

Comment#1: The authors didn’t report the stability comparison between the optimized metal- sulfide and oxide layers. So, it is not clear how the oxidation process affects the stabilities of the layered materials. Consider providing a comparison of formation energies of metal-sulfide and oxide layers studied in this work.

Comment#2: Consider updating Figure 3 like Figure 4/5/6.

Comment#3: Figure 4. Correct the labeling of Ta/Sn/Ta. Remove the space between T and a, and the same for others. Why are the boxes tilted randomly?

Comment#4: Why are Au and Al selectively chosen as electrodes?

Author Response

The response to reviewer 2 is attached.

Reviewer 3 Report

Comments and Suggestions for Authors

Author Response

The response to reviewer 3 is attached.

Round 2

Reviewer 3 Report

Comments and Suggestions for Authors

The authors have addressed all issues raised in my original report and I am happy with the revised manuscript.

Comments on the Quality of English Language

The English requires some careful editing.